

# Enhanced futures price-spread forecasting based on an attention-driven optimized LSTM network: integrating an improved grey wolf optimizer algorithm for enhanced accuracy

Yongli Tang[1], Zhenlun Gao[1], Zhongqi Cai[1], Jinxia Yu[2] and Panke Qin[1]

[1] School of Software, Henan Polytechnic University, Jiaozuo, Henan, China
[2] School of Computer Science and Technology, Henan Polytechnic University, Jiaozuo, Henan, China

## ABSTRACT

Financial market prediction faces significant challenges due to the complex temporal dependencies and heterogeneous data relationships inherent in futures price-spread data. Traditional machine learning methods struggle to effectively mine these patterns, while conventional long short-term memory (LSTM) models lack focused feature prioritization and suffer from suboptimal hyperparameter selection. This article proposes the Improved Grey Wolf Optimizer with Multi-headed Self-attention and LSTM (IGML) model, which integrates a multi-head self-attention mechanism to enhance feature interaction and introduces an improved grey wolf optimizer (IGWO) with four strategic enhancements for automated hyperparameter tuning. Benchmark tests on optimization problems validate IGWO's superior convergence efficiency. Evaluated on real futures price-spread datasets, the IGML reduces mean square error (RMSE) and mean absolute error (MAE) by up to 88% and 85%, respectively, compared to baseline models, demonstrating its practical efficacy in capturing intricate financial market dynamics.

## INTRODUCTION

The prosperity of every developing economy, nation, or community in the 21st century depends on their market economies, particularly with the financial market serving as the central point (*Nassirtoussi et al., 2014*). Consequently, an extensive study and understanding of the financial market are both crucial and indispensable. Predicting financial markets is extremely challenging due to uncertainties such as overall economic conditions at domestic and international levels, social factors, and political events (*Zhao et al., 2023*; *Song et al., 2023*; *Kehinde, Chan & Chung, 2023*). However, compared to other high-risk, high-reward financial derivatives, futures price-spread arbitrage can achieve a low-risk, high-reward profile. Additionally, futures feature a two-way trading mechanism, high leverage, flexible trading hours, and direct market responses. These characteristics provide the futures market with more trading opportunities and the potential for higher

Corresponding author
Panke Qin, qinpanke@hpu.edu.cn

returns compared to the financial derivatives market. Hence, futures trading plays a pivotal role in the economic and financial sphere, exerting influence on both banking institutions and the market for financial derivatives (*Su et al., 2023*; *Mohsin & Jamaani, 2023*; *He et al., 2023*). Historically, futures spread arbitrage has appealed to numerous financial experts due to its supportive capacity and consistent returns. It is worth noting that even a slight fluctuation in the price spread of certain commodities can lead to substantial profits or, conversely, considerable investment and economic losses. The trend of futures prices is influenced by numerous factors, such as supply-demand dynamics, seasonal variations, natural calamities, and policy fluctuations. Additionally, there is a potential connection between the prices of different commodities. For example, if the future price of raw materials drops significantly, the prices of manufactured goods for processing enterprises may also decrease. Such price changes can be profitable for investors (*Lang, 1995*). Consequently, forecasting commodity price spreads can aid financial investors in formulating sound investment strategies and minimizing potential risks. However, the inherent complexity of accurately predicting price-spread trends is widely regarded as a highly demanding and difficult endeavor (*Li & Song, 2023*; *Deng et al., 2023*; *Cheung et al., 2023*).

Traditional finance has formed an abstract theoretical framework composed of theories such as Portfolio Theory (PT), Capital Asset Pricing Model (CAPM), Arbitrage Pricing Theory (APT), Efficient Market Hypothesis (EMH), and the Black-Scholes (BS) option pricing model. The assumptions of mainstream economics are based on absolute rationality, yet human decision-making inevitably involves emotional factors. Hence, with the advancement of machine learning, investors have embarked on endeavors to apply it within the realm of finance to assist them in making rational decisions. For decades, research has been conducted on forecasting futures price-spread trends and analyzing its influencing factors, resulting in various proposed approaches. Traditional time-series methods, including multiple linear regression, along with the widely recognized Auto-Regressive Integrated Moving Average (ARIMA) model, have been employed to address the challenge of futures price-spread prediction (*Ji et al., 2019*; *Wang & Zhang, 2020*). In *Ray et al. (2023)*, proposed an enhanced hybrid ARIMA-long short-term memory (LSTM) model that leverages a random forest for lag selection. The ARIMA component estimates the mean effect, while the Generalized Autoregressive Conditional Heteroskedasticity (GARCH) model, applied to the ARIMA residuals, captures the volatility of the series. Alongside traditional econometric and time-series methods, machine learning techniques have been deployed to uncover the intricate patterns within futures prices (*Li & Song, 2023*; *Zhao, 2021*; *Singh et al., 2023*). *Kuo & Chiu (2024)*, in article, devised a novel predictive model that integrates jellyfish search and particle swarm optimization (HJPSO) to fine-tune support vector machine (SVM) parameters, with SVM's classification capabilities assessed through comparative experiments. Meanwhile, financial researchers employed traditional statistical methods and signal processing techniques to analyze stock market data. However, both statistical methods and machine learning approaches often struggle to capture and model the nonlinear and complex

dynamics within futures price-spread time series. Therefore, these methods are all inadequate for the task of futures price-spread prediction.

In recent years, deep learning has been increasingly applied in various fields, including time-series forecasting (*Wang et al., 2023*; *Lin et al., 2023*; *Stefenon et al., 2023*; *Kim, Kang & Kang, 2023*). These technologies have demonstrated high performance when dealing with nonlinear and highly volatile time series. Among the most popular, efficient, and widely utilized deep learning approaches are long short-term memory (LSTM) networks and convolutional neural networks (CNNs). *Li, Guan & Liu (2023)* in article proposed a model based on CNN and LSTM neural networks to account for spatiotemporal correlations and external features in flight delay prediction. The CNN is utilized to learn spatial correlations, while the LSTM captures temporal correlations to enhance prediction accuracy. *Wu et al. (2023)* used the array as the input image of the CNN framework, extracting certain feature vectors through the convolutional layer and the pooling layer, and using them as the input vector of LSTM. The fundamental concept behind using these models for time-series problems lies in the fact that LSTM models, thanks to their unique architecture, are adept at capturing sequential pattern information. Conversely, CNN models excel at filtering out noise from input data and extracting valuable features that significantly contribute to the final prediction model. Although LSTM networks are specifically designed to handle temporal correlations, they tend to exploit only the features available in the training set. To overcome this issue, integrating the strengths of different deep learning techniques can enhance the predictive performance of time-series models. However, integrating multiple models introduces the challenge of selecting additional hyperparameters.

Metaheuristic algorithms are strategy-based techniques that intelligently explore the search space of optimization problems to discover solutions that are close to optimal. Consequently, metaheuristics can be employed to optimize the hyperparameters of LSTM models, offering greater efficiency than traditional optimization algorithms. *Ren et al. (2021)* in article utilized the Squirrel Search algorithm to optimize the weights of the neural network, applying the optimized model to stock price prediction. *Guo & Wang (2024)* utilized genetic algorithm and particle swarm optimization to optimize the Back Propagation (BP) neural network model, achieving promising results in stock price prediction as well. Experiments demonstrated that the proposed method achieves accurate estimations under various conditions. The advantage of metaheuristic algorithms lies in their ability to explore a larger search space and efficiently find the optimal global solution, making them highly suitable for optimizing the numerous hyperparameters in neural networks. For instance, when dealing with complex models such as LSTM networks, traditional grid search or random search methods may be inefficient in handling such a vast number of hyperparameters. In contrast, metaheuristic algorithms can leverage their intelligent search strategies to find superior combinations of hyperparameters in a shorter amount of time, thereby enhancing the performance and accuracy of neural networks. However, even in these demanding scenarios, metaheuristic algorithms can manage this complexity and noise to identify the best hyperparameters (*Gülmez, 2023*). However,

traditional metaheuristic algorithms often fall into local optima, thereby failing to find the global optimal solutions.

Existing approaches to futures price-spread forecasting exhibit three critical limitations. First, conventional machine learning models fail to capture nonlinear interdependencies between heterogeneous data sources, leading to oversimplified representations of market microstructure. Second, while LSTM-based methods partially address temporal dependency modeling, their fixed window-based processing often overlooks irregularly spaced critical events, and their monolithic attention mechanisms struggle to prioritize concurrent multi-scale market signals. Third, hyperparameter optimization in current frameworks predominantly relies on grid search or vanilla evolutionary algorithms, which lack the efficiency to handle high-dimensional parameter spaces in dynamic trading environments—resulting in models that rapidly become obsolete during market regime shifts. These shortcomings collectively hinder real-world deployment, particularly in scenarios requiring adaptive responses to volatile, high-frequency data streams.

To tackle the aforementioned issues, this article proposes a hybrid forecasting model that leverages the strengths of deep learning techniques. The model employs a multi-head self-attention mechanism to capture extensive dependencies and learn the intrinsic representations of time-series data. Additionally, an enhanced Grey Wolf Optimizer is introduced in this article to dynamically optimize the model parameters, thereby enhancing the model's predictive capabilities. By analyzing historical futures price-spread data, the model is able to predict the price-spread at the next time point and continuously improve its predictive accuracy through the refined GWO.

This work makes the following key contributions:

(1) To tackle the issue of the GWO easily getting trapped in local optima, four enhancement methods are proposed in this article. Through rigorous testing using Congress on Evolutionary Computation (CEC) benchmark functions, the superiority of the enhanced algorithm is demonstrated.

(2) To address the challenge of model parameter selection, this study integrates swarm optimization algorithms to dynamically determine appropriate parameters for the model under varying conditions, leading to a significant enhancement in the model's predictive capabilities. The improved model is abbreviated as IGML (Improved Grey Wolf Optimizer with Multi-headed Self-attention and LSTM) in the following text.

(3) To mitigate the problem of gradient explosion in LSTM models, a multi-head self-attention mechanism is introduced in this article. Experimental results indicate improvements in key performance metrics such as root mean square error (RMSE), mean absolute error (MAE), and coefficient of determination ($R^2$), validating the efficacy of the proposed approach.

The remainder of this article is divided into several sections: "Methodology" introduces in detail the proposed deep learning model and the improved metaheuristic algorithm. "Experiments" describes the data collection process and the simulation experiments conducted. "Conclusion" presents the experimental conclusions and proposes some ideas for improvement.

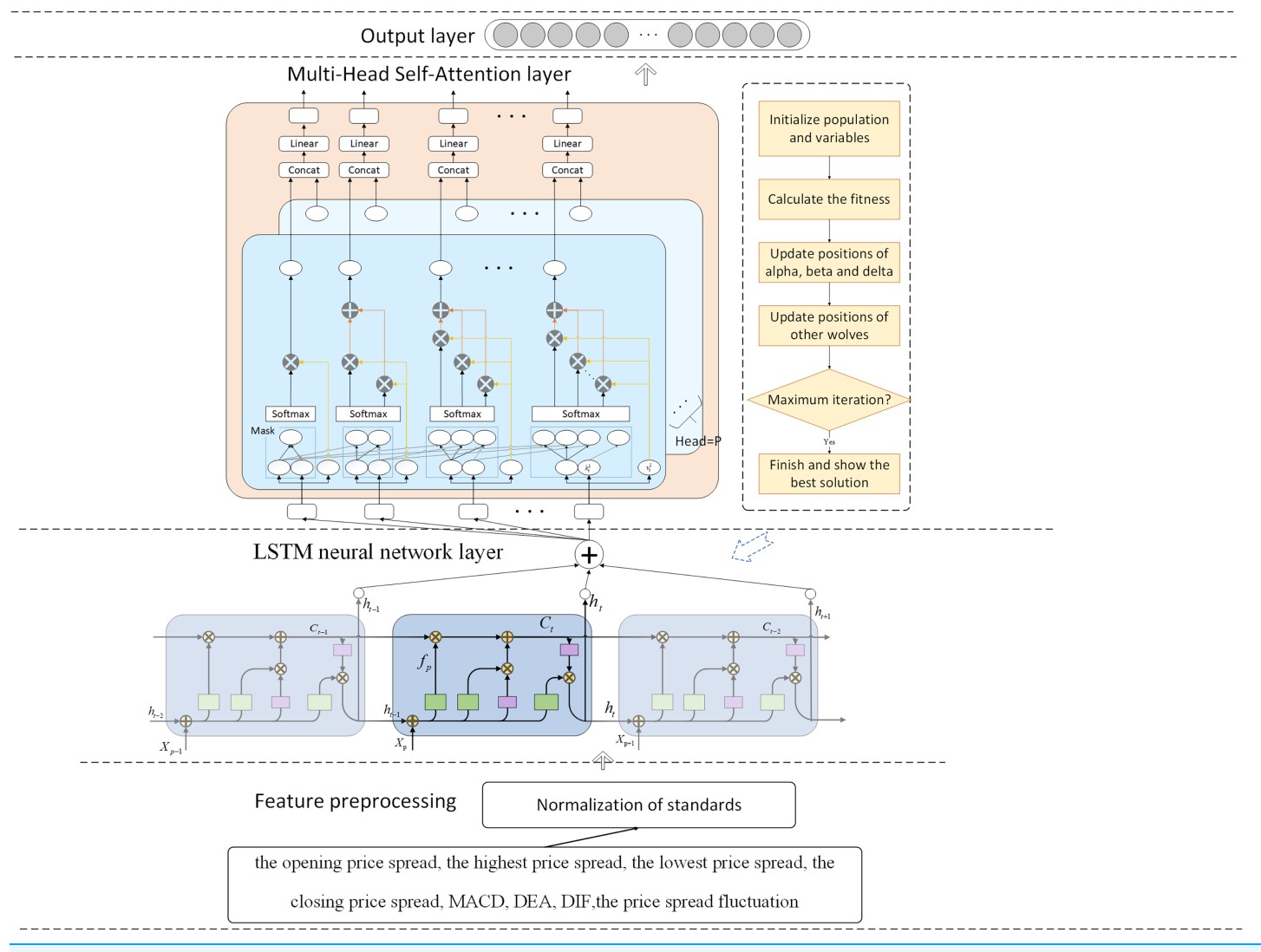

**Figure 1 Framework of proposed method for futures price-spread forecasting.**

# METHODOLOGY

## IGML model

The research presented an IGWO-LSTM network and a multi-head self-attention mechanism for futures price-spread forecasting. Figure 1 depicts the futures price-spread forecasting framework.

The prediction model comprises three pivotal components: the feature decomposition layer, the LSTM neural network layer, and the attention mechanism layer. Additionally, the parameters of the LSTM network layer are optimized using the Improved Grey Wolf Optimizer.

In this framework, data features are first preprocessed and selected, and then the processed features are input into the LSTM for training and prediction. To capture the relationships between features and highlight important features, a multi-head

self-attention mechanism layer is added after the LSTM layer. Finally, the prediction results are output through a fully connected layer. To select appropriate hyperparameters for the model, the GWO is introduced for parameter optimization. To address the issues of slow convergence and susceptibility to local optima in GWO, this article proposes four improvement measures.

The IGML model, which integrates LSTM with a multi-head self-attention mechanism and utilizes IGWO, exhibits significant advantages in handling complex temporal dependencies in financial data. The LSTM network is inherently adept at capturing long-term dependencies in sequential data, effectively addressing the gradient vanishing and gradient explosion issues faced by traditional recurrent neural networks (RNNs) when processing long sequences through its unique gating mechanisms. The incorporation of the multi-head self-attention mechanism further enhances the model's ability to focus on information from different positions in the sequence, enabling it to precisely capture nuanced and crucial temporal dependency features in financial data. Additionally, the application of IGWO for hyperparameter tuning optimizes key parameters within the LSTM and multi-head self-attention mechanisms, further improving the model's generalization ability and prediction accuracy.

The subsequent discourse furnishes an elaborate exposition of each constituent element comprising the predictive model.

## Proposed model

### Long short-term memory network

The LSTM neural network, introduced by *Hochreiter & Schmidhuber (1997)*, is RNN featuring nonlinear gated units and memory cells that facilitate the capture of long-term dependencies in sequential data (*Greff et al., 2016*). This capability is achieved through selective information processing mechanisms, including reading, writing, and forgetting, which ensure that only pertinent information is retained within the memory cells.

The LSTM network regulates the flow of information through its memory cells using memory cells and three gates. The specific structure is illustrated in Fig. 2.

The mathematical formulas for various operations performed in LSTM are shown in Eqs. (1) to (8).

$$f_t = \sigma\left(W_f X_t + U_f h_{t-1} + b_f\right) \tag{1}$$

$$\hat{C}_t = \tanh \leq \left(W_c X_t + U_c h_{t-1} + b_c\right) \tag{2}$$

$$i_t = \sigma(W_i X_t + U_i h_{t-1} + b_i) \tag{3}$$

$$C_t = f_t \times C_{t-1} + i_t \times \hat{C}_t \tag{4}$$

$$O_t = \sigma(W_o X_t + U_o h_{t-1} + b_o) \tag{5}$$

$$h_t = O_t \times \tanh(C_t) \tag{6}$$

$$\sigma(x) = \frac{1}{(1 + e^{-x})} \tag{7}$$

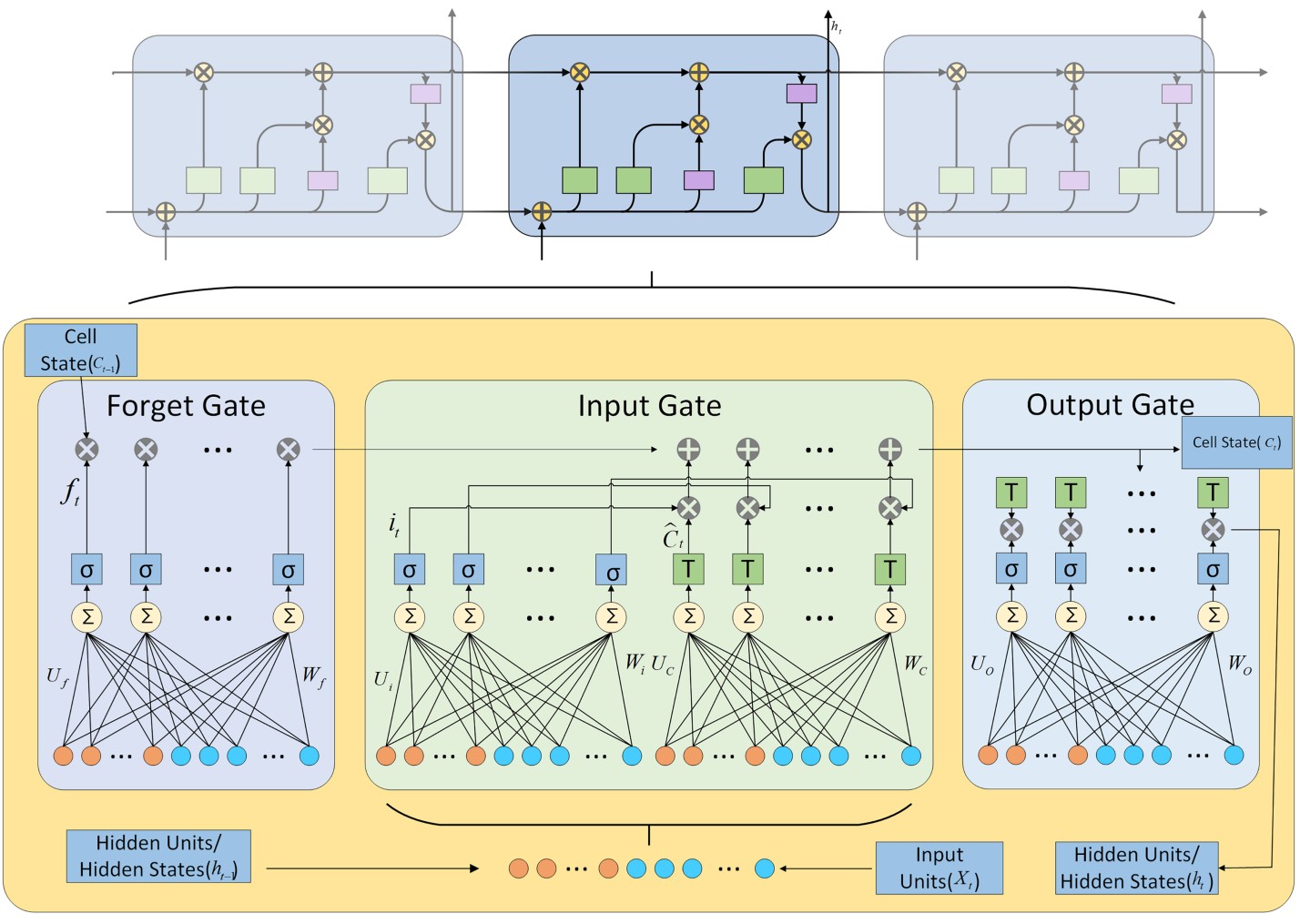

**Figure 2 LSTM architecture diagram.**

$$\tanh(x) = \frac{(e^x - e^{-x})}{(e^x + e^{-x})} \tag{8}$$

$\sigma(x)$ is log-sigmoid activation function and $\tanh(x)$ is the hyperbolic tangent activation function.

### Multi-head self-attention

To capture the relationships between features and highlight important information, this article introduces a multi-head self-attention mechanism within a three-layer LSTM network. *Zhou et al. (2025)* integrated large language models (LLM), linear transformers (LT), and convolutional neural networks to enhance stock price prediction using historical market data. By integrating multiple self-attention modules that perform feature learning in distinct state subspaces, the multi-head structure significantly enhances the model's ability to learn long-term dependencies. Essentially, multi-head attention (MHA) achieves

superior prediction performance through the interplay among its individual heads, whereas a single-head structure tends to focus solely on important features at certain stages, neglecting others. Compared with traditional recurrent layers, self-attention can process all elements in a sequence in parallel and directly capture dependencies between any two elements in the sequence, which helps the model to more accurately understand the global structure of the data. Furthermore, by calculating relevance weights between different elements, self-attention dynamically adjusts the level of attention to different information, enhancing the model's flexibility and expressive power.

By designating $\theta = \{\theta_1, ..., \theta_t\}^T \in R^{t \times d_{em}}$ as the input sequence and $B = \{b_1, ..., b_t\}^T \in R^{t \times d_{em}}$ as the output sequence, the configuration of the masked Multi-Head Attention (MHA) module is depicted in Fig. 3. The computational steps involved in the masked MHA are outlined in Eqs. (9) to (13).

$$q_i^h = (\theta_i)(W^q)^h k_i^h = (\theta_i)\left(W^k\right)^h v_i^h = (\theta_i)(W^v)^h \left(q_i^h, k_i^h, v_i^h \in R^{1 \times (d_k)}\right) \tag{9}$$

$$a_{i,\tau}^h = \frac{q_i^h \cdot \left(k_i^h\right)^T}{\sqrt{d_k}} \tag{10}$$

$$b_i^h = \sum_{\tau=1}^{i} Soft \max \left(a_{i,\tau}^h\right) v_i^h \tag{11}$$

$$b_i^h = Soft \max \left(\frac{Q^h \left(K^h\right)^T}{\sqrt{d_k}}\right) V^h (h \in \{1, ..., P\}) \tag{12}$$

$$b_i = linear\left\{concat\left(b_i^1, b_i^2, ..., b_i^P\right)\right\} = concat\left(b_i^1, b_i^2, ..., b_i^P\right) W^o. \tag{13}$$

The attention mechanism allocates varied weights to input features, thereby emphasizing crucial factors, mitigating the impact of weakly correlated factors, capturing correlations between multiple variables and prediction outcomes, and determining long-range dependencies within the input sequence. This approach optimizes the network structure. In contrast to traditional factor selection and uniform weight assignment, the attention mechanism enhances prediction performance by emphasizing important information through strategic weight allocation.

## Algorithmic approaches and enhancements

### Grey wolf optimizer

To tackle the problem of selecting optimal model parameters, *Emary, Zawbaa & Grosan (2017)* proposes the GWO algorithm, which mimics the natural behavior of grey wolves. The GWO algorithm was selected as the core methodology in this study due to its unique bio-inspired mechanisms and efficient optimization performance. By simulating the social hierarchy and cooperative hunting behavior (encompassing tracking, encircling, and attacking phases) of grey wolf packs, this algorithm inherently achieves dynamic equilibrium between global exploration and local exploitation. The leadership hierarchy guides the population toward potential optimal regions, while subordinate wolves maintain search diversity through information sharing, effectively mitigating premature

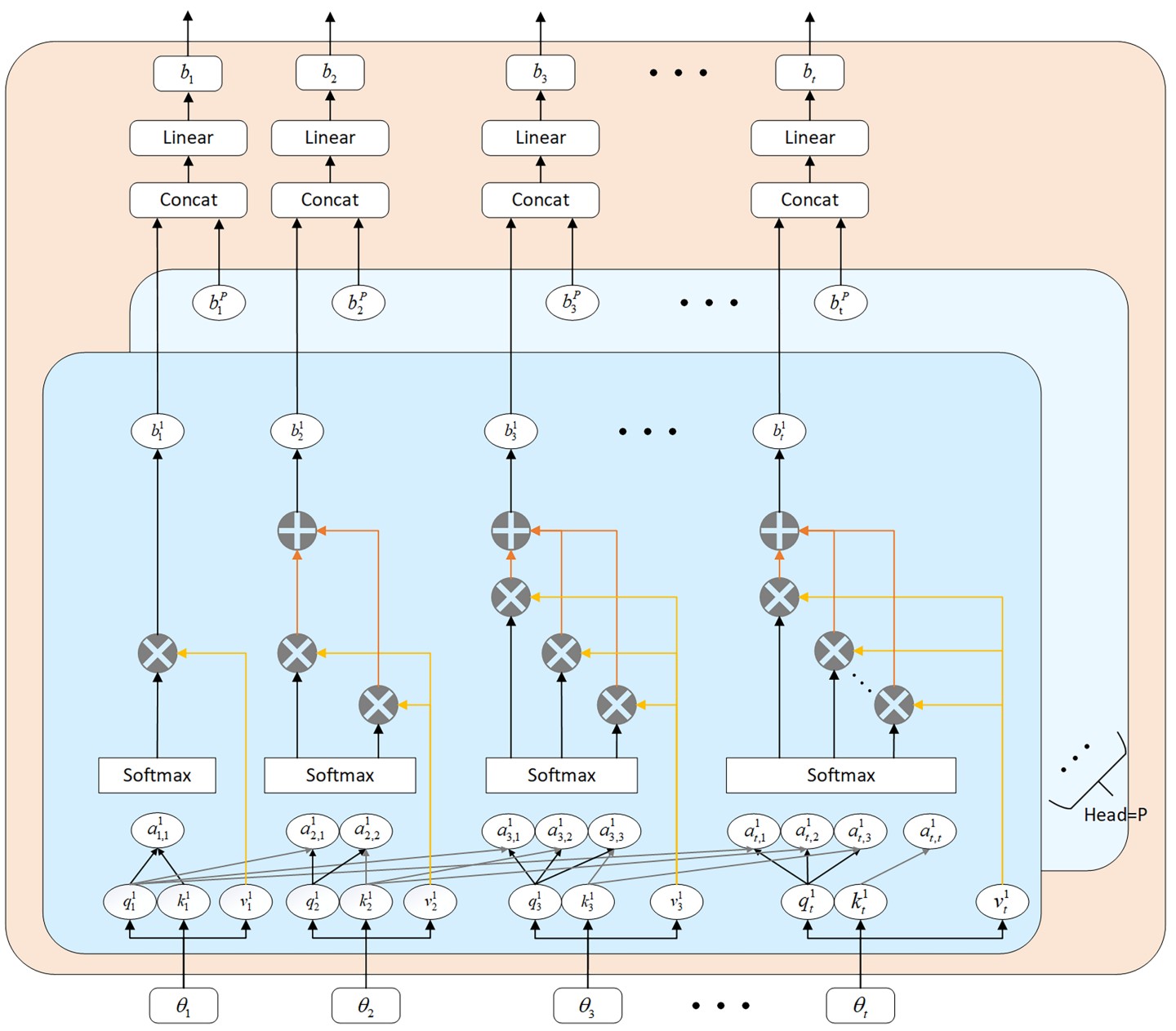

**Figure 3 Multi-head self-attention mechanism architecture diagram.**

convergence in GA and local optima entrapment in PSO. Compared with conventional algorithms, GWO requires only the adjustment of population size and iteration count, exhibiting a streamlined parameter configuration that substantially reduces tuning complexity. Furthermore, experimental results demonstrate that GWO achieves accelerated convergence rates and enhanced solution precision in high-dimensional, nonlinear optimization problems. Its hierarchical collaboration mechanism proves particularly advantageous for multimodal function optimization scenarios, thereby

providing a robust and efficient solution framework for addressing complex engineering challenges. The mathematical formulas are presented in Eqs. (14) to (21).

$$D = |C \times X_P - X(t)| \tag{14}$$

$$X(t+1) = X_P(t) - A \times D \tag{15}$$

$$A = 2 \times A \times r_1 - a(t) \tag{16}$$

$$C = 2 \times r_2 \tag{17}$$

$$a(t) = 2 - \frac{(2 \times t)}{MaxIter} \tag{18}$$

$$\begin{aligned} D_\alpha &= |C_1 \times X_\alpha - X(t)| \\ D_\beta &= |C_2 \times X_\beta - X(t)| \\ D_\delta &= |C_3 \times X_\delta - X(t)| \end{aligned} \tag{19}$$

$$\begin{aligned} X_{i1}(t) &= X_\alpha(t) - A_{i1} \times D_\alpha(t) \\ X_{i2}(t) &= X_\beta(t) - A_{i2} \times D_\beta(t) \\ X_{i3}(t) &= X_\delta(t) - A_{i3} \times D_\delta(t) \end{aligned} \tag{20}$$

$$X(t+1) = \frac{X_{i1}(t) + X_{i2}(t) + X_{i3}(t)}{3}. \tag{21}$$

While the GWO algorithm is straightforward and suitable for various applications, it faces challenges such as limited population diversity, an uneven balance between exploitation and exploration, and premature convergence (*Heidari & Pahlavani, 2017*). Additionally, the position update equation in GWO excels at exploitation but falls short in achieving feasible solutions. To overcome these limitations, this article introduces an enhancement strategy.

### Slime mold algorithm

To address the tendency of GWO to converge prematurely to local optima, this article proposes a hybrid approach combining GWO with the Slime Mold Algorithm (SMA). SMA is a new meta-heuristic algorithm proposed by *Li et al. (2020)*. Compared to other intelligent optimization algorithms, slime mold algorithm has advantages such as simple principles, ease of implementation, few adjustable parameters, and strong optimization capabilities.

All individuals within the swarm are initialized randomly and uniformly across the entire domain (LB, UB):

$$X_i = r_1(UB - LB) + LB \tag{22}$$

$$X_i(t+1) = \begin{cases} r_2 \times (UB - LB) + LB, & r_2 < z \\ X_b + v_b \times [W \times X_A(t) - X_B(t)], & r_3 < p \\ v_c \times X_i(t), & p \leq r_3 \leq 1 \end{cases} \tag{23}$$

$$a = \tanh\left(1 - \frac{t}{Max\_iter}\right) \tag{24}$$

$$b = 1 - \frac{t}{Max\_iter} \tag{25}$$

$$p = \tanh|S_i - DF| \tag{26}$$

$$W_{si}(i) = \begin{cases} 1 + r_4 \times \log\left(1 + \dfrac{bF - S_i}{bF - wF}\right), & condition \\ 1 - r_4 \times \log\left(1 + \dfrac{bF - S_i}{bF - wF}\right), & others \end{cases} \tag{27}$$

$$S_i = sort(S). \tag{28}$$

### Improvement strategy

To enhance the global search capability of GWO and accelerate its convergence speed, this article proposes the following three improvement strategies.

(1) NSGA-II improved population initialization

NSGA-II was originally proposed by *Deb et al. (2002)*, with the basic definition and construction:

Suppose $G_s$ is a s-dimensional Euclidean space, where $r \in G_s$, then $P_n(i) = (r_1 i_1, r_2 i_2, r_3 i_3, \ldots, r_n i_n), i = 1, 2, 3, \ldots, n$. $n$ represents the sample size, $P_n(i)$ represents the set of non-dominated solutions, and $r$ refers to a non-dominated solution, usually taken as $r = \left\{2 \cos \frac{2\pi j}{7} i, 1 \leq i \leq n; 1 \leq j \leq s\right\}$ or $r = \{e^j i\}$. Here, J is the smallest prime number satisfying $(k - 3)/2 \geq 0$.

Step 1: Calculate $r$ value, $r = (r_1, r_2, r_3, \ldots, r_n)$, where $r_j = \left(2 \cos\left(\frac{2\pi j}{7}\right) m_i, 1\right), 1 \leq j \leq n$. $n$ represents the dimension, $m$ represents the population size, and $m_i$ represents the i-th individual.

Step 2: Construct a set of optimal points with a quantity of $m$: $P_n(i) = \{(r_1 i_1, r_2 i_2, \ldots, r_n i_n)\}, i = 1, 2, 3, \ldots n$

Step 3: Map $P_n$ to the feasible domain where the population resides: $X_i^j = a_j + P_n(i)(b_j - a_j)$. Here, $a_j$ represents the lower bound of the current dimension, and $b_j$ represents the upper bound of the current dimension.

Assuming a population size of 100, the comparison between the initialization of the optimal point set population and the random initialization population is shown in the Fig. 4.

(2) Nonlinear convergence factor variation

In vanilla GWO, when $|A| > 1$, the wolf population searches for potential prey throughout the entire search domain; when $|A| < 1$, the wolf population gradually surrounds and captures the prey. The value of $A$ depends on the variation of the convergence factor.

In *Rodríguez et al.*'s *(2017)* study, it has been demonstrated that different updating strategies for the critical parameter $a$ can greatly impact the algorithm's performance, and linear strategies are often not the most effective. Therefore, this article proposes a new

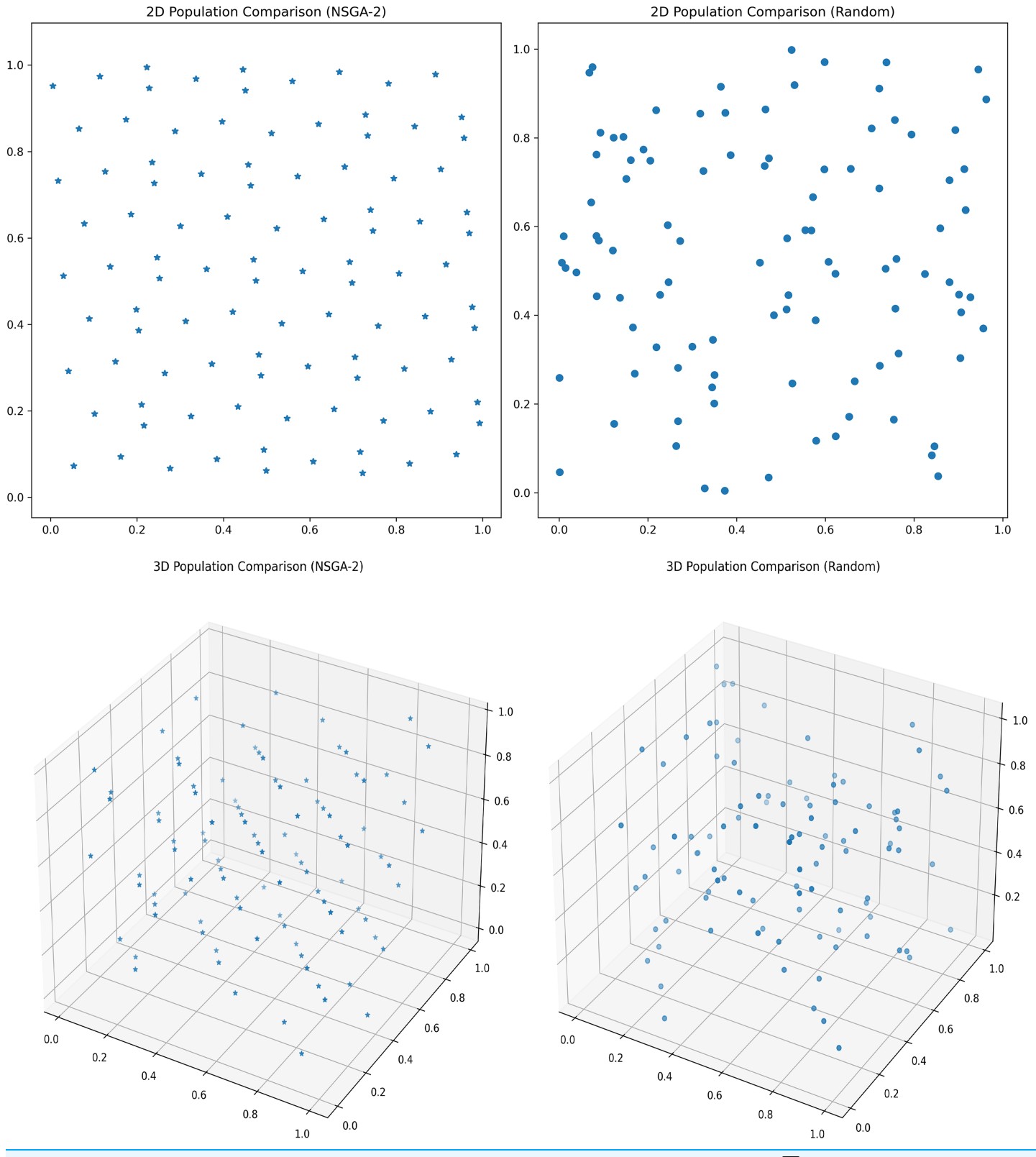

**Figure 4 Initial population distribution.**

convergence factor updating method based on the periodic variation of the trigonometric function, as shown in Eq. (29):

$$a = 1 + \sin\left(\left(\frac{\pi}{2}\right) + \pi \times \left(\frac{l}{Max\_iter}\right)\right) \tag{29}$$

As shown in the Fig. 5, the straight line represents the convergence factor updating method used in GWO, while the curved line represents the new convergence factor updating method proposed in this article.

The graph of the improved convergence factor is a curve based on trigonometric function variation. It decreases slowly in the early iterations, allowing the convergence factor to maintain a relatively large value for a longer time, thus enhancing search efficiency. In the later iterations, it decreases rapidly, allowing the convergence factor to maintain a relatively small value for an extended period, thereby improving search accuracy. Therefore, the algorithm can focus on specific behaviors at different stages.
(3) Dynamic updating of leaders' weights

In *Zhang & Zhou (2021)*, the shortcomings of the GWO position update formula are pointed out, where the averaging of $X_1$, $X_2$, and $X_3$ values fails to highlight the importance of $\alpha$, $\beta$, and $\delta$. To address this issue, *Chiu, Shih & Li (2018)* introduces three improvement strategies into the algorithm: the exponential law for adjusting the convergence factor, the adaptive position updating strategy for grey wolves, and the revised dynamic weight strategy.

The current distance weights of individual grey wolves to $\alpha$, $\beta$, and $\delta$ are given by Eqs. (30) to (32).

$$W_1 = \frac{|X_1|}{|X_1| + |X_2| + |X_3|} \tag{30}$$

$$W_2 = \frac{|X_2|}{|X_1| + |X_2| + |X_3|} \tag{31}$$

$$W_3 = \frac{|X_3|}{|X_1| + |X_2| + |X_3|}. \tag{32}$$

However, in practical applications, Eqs. (30) to (32) may result in a denominator of zero. Therefore, it is necessary to add a very small constant $\varepsilon$, with a value of $e^{-16}$. The modified equations are shown in Eqs. (33) to (35):

$$W_1 = \frac{|X_1|}{|X_1| + |X_2| + |X_3| + \varepsilon} \tag{33}$$

$$W_2 = \frac{|X_2|}{|X_1| + |X_2| + |X_3| + \varepsilon} \tag{34}$$

$$W_3 = \frac{|X_3|}{|X_1| + |X_2| + |X_3| + \varepsilon}. \tag{35}$$

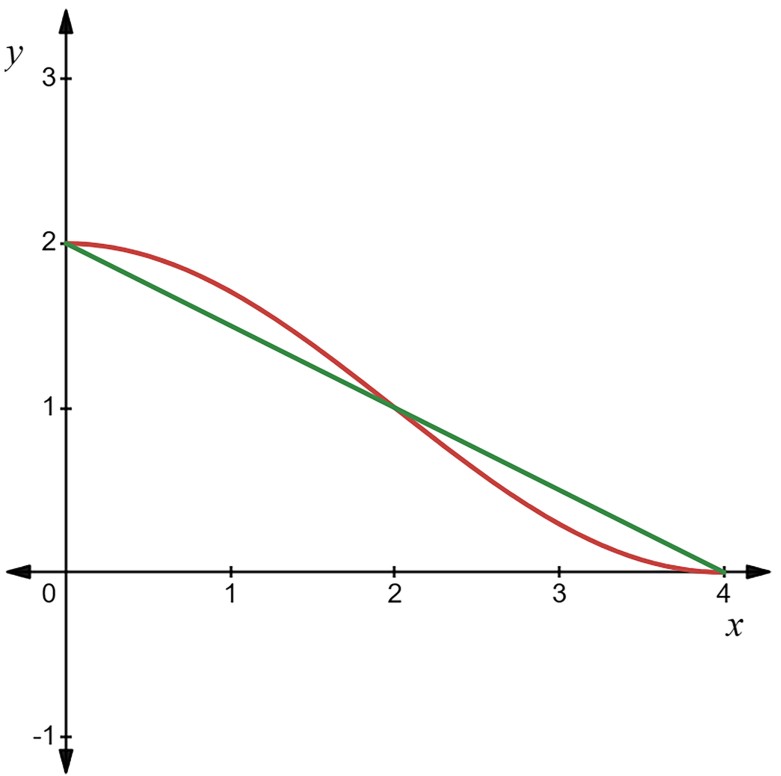

**Figure 5 Comparison of convergence factors variations.**

Combining the adaptive position update strategy, the final grey wolf position update method is given by Eq. (34):

$$X(t+1) = \frac{W_1X_1 + W_2X_2 + W_3X_3}{3} \times \left(1 - \frac{t}{T}\right) + X_1 \times \frac{t}{T}. \tag{36}$$

### *Improved grey wolf optimizer*

By hybridizing the GWO and SMA, we combine the optimal position update formulas of both methods to yield the complete update formula. For individuals ranking high in terms of fitness, the updating equation for the hybrid GWO-SMA algorithm would be:

$$X_i(t+1) = \begin{cases} X_b + v_b \times [W \times X_A(t) - X_B(t)], r_c \leq 0.5 \\ \frac{W_1X_1 + W_2X_2 + W_3X_3}{3} \times \left(1 - \frac{t}{T}\right) + X_1 \times \frac{t}{T}, r_c > 0.5 \end{cases}. \tag{37}$$

For wolves ranking low in fitness, the update formula is as Eq. (38):

$$X_i(t+1) = \begin{cases} X_b + v_b \times [W \times X_A(t) - X_B(t)], r_c \leq 0.25 \\ \frac{W_1X_1 + W_2X_2 + W_3X_3}{3} \times \left(1 - \frac{t}{T}\right) + X_1 \times \frac{t}{T}, 0.25 < r_c \leq 0.5 \\ r_2 \times (UB - LB) + LB, 0.5 < r_c \leq 1 \end{cases}. \tag{38}$$

The complete flowchart of the algorithm is shown as Fig. 6.
The algorithm pseudocode is as Box 1 shows.

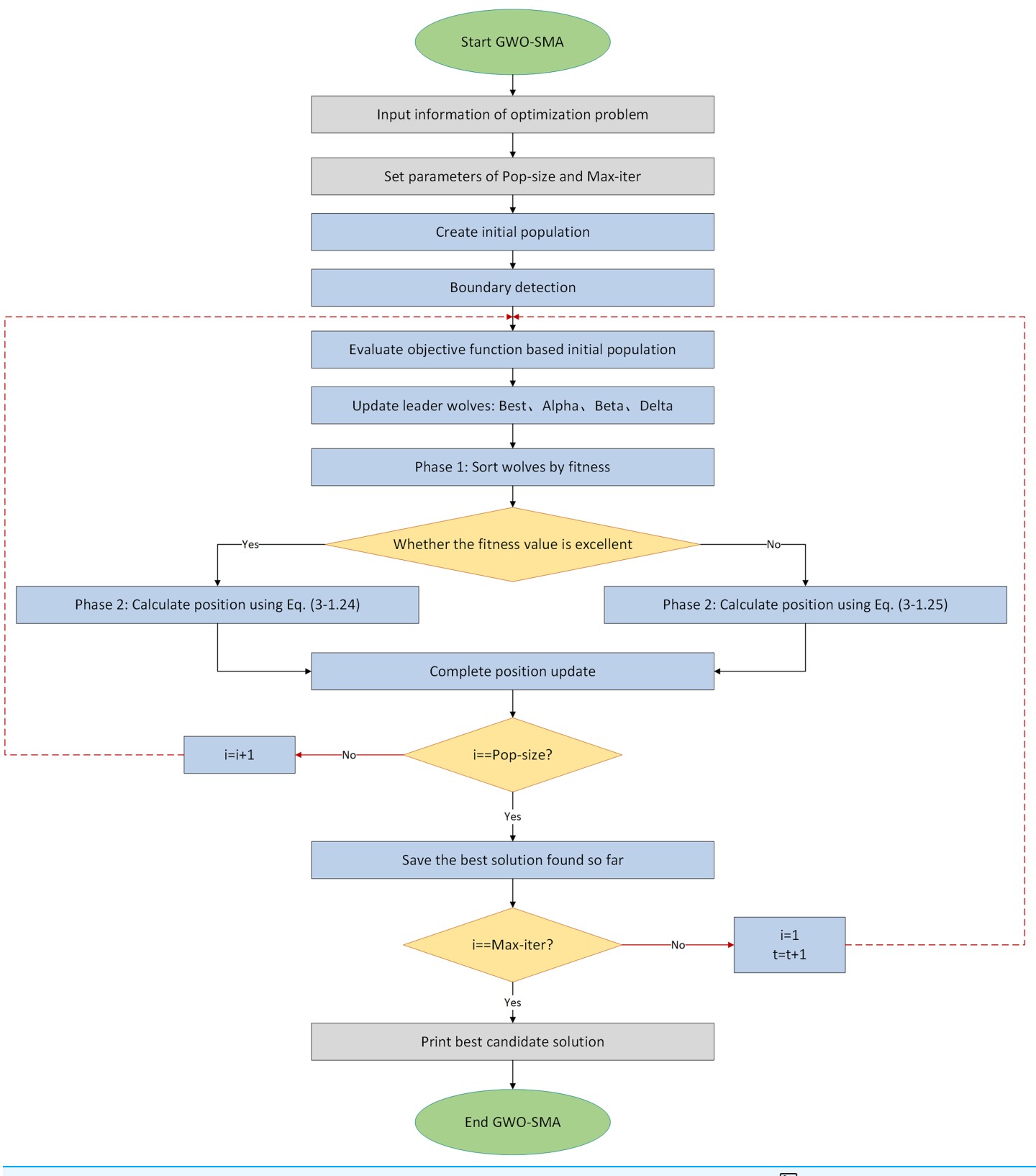

**Figure 6  Flowchart of IGWO algorithm.**

By incorporating several key improvements, the performance of IGWO has been enhanced in terms of efficiency, convergence speed, and the ability to avoid local optima, making it a more robust and versatile optimization tool compared to the standard GWO.

## EXPERIMENTS

### Data preparation

In this section, we offer a clear and concise overview of the data employed in our study, validating its relevance and suitability through correlation analysis and the Engle-Granger (EG) cointegration test.

The data was sourced from the Shanghai Futures Exchange in China, which utilizes the CTP protocol to provide snapshot-based order data aggregated over 500-ms intervals. The original data used in the analysis originated from Shanghai Futures Exchange, and tick data was exported through Python for fitting. By leveraging these tick data from rebar and hot-rolled coil contracts, we computed the spread information, yielding one-minute K-line data. The final dataset spans from 21:01 on July 15, 2020, to 10:50 on March 23, 2023, encompassing 225,155 data points across 654 days.

The price spread data for each period (1 min) encompasses eight features: the opening price spread, highest price spread, lowest price spread, closing price spread, MACD, DEA, DIF, and price spread fluctuation. Among these, the closing price spread serves as our prediction target. The dataset is divided into a training set (80%), a test set (20%), and a validation set (10% of the training set). The training set is used to train the model, teaching it to map input features to targets. The validation set assesses the model's performance during training and aids in hyperparameter tuning. Finally, the test set evaluates the model's overall performance.

To confirm whether there exists a long-term and stable cointegration relationship between the selected futures contracts, Therefore, we conducted a cointegration analysis on the closing prices of rebar and hot-rolled coil (HRC) futures.

Initially, unit root analysis was performed on both futures contracts' closing prices to assess stationarity characteristics. As evidenced in Table 1, the logarithmic price series for both instruments exhibited non-stationary properties with the presence of unit roots. Subsequent examination of their first-order differenced series revealed stationarity, as indicated by the rejection of unit root null hypotheses in differenced data.

An econometric model was subsequently established to examine the relationship between Hot Rolled Coil Futures (HC) and Ribbed Bar Futures (RB) closing prices. Post-estimation diagnostic checks focused on the model's residuals (resid01) through additional unit root testing. Table 2 demonstrates that the residual series achieved stationarity at conventional significance levels. This empirical evidence—non-stationary original series coupled with stationary residuals and stationary differenced series—satisfies the statistical requirements for cointegration. The results therefore confirm the existence of a stable long-term equilibrium relationship between HC and RB futures prices, consistent with cointegration theory.

**Table 1 Unit root test.**

| Variety | Dickey-Fuller | *P*-value | Stationarity |
|---------|---------------|-----------|--------------|
| HC | 0.035680 | 0.6942 | False |
| RB | −0.025008 | 0.6745 | False |
| ΔHC | −96.58328 | 0.0001 | True |
| ΔRB | −93.88820 | 0.0001 | True |

Note:
   HC, Hot Rolled Coil Futures; RB, Ribbed Bar Futures.

**Table 2 Residual series test.**

| Residual | ADF statistic | *P*-value | Result |
|----------|---------------|-----------|--------|
| Resid01 | −7.235169 | 0.0000 | Stationary |

Note:
   ADF, Augmented Dickey-Fuller test.

## Statistics

### CEC 2019 benchmark

Each algorithm was individually assessed on the benchmark set to measure its performance on specific test cases. The detailed parameters of the CEC2019 function set are presented in Table 3. It mainly includes multimodal optimization problems, high-dimensional optimization problems, unimodal and multimodal functions, hybrid and composite functions, and other types of optimization problems.

### Statistical results

Table 4 presents the statistical comparison of IGWO against other benchmark algorithms on the CEC 2019 benchmark functions. In terms of average performance, IGWO achieves the top rank on eight functions (CEC-01, CEC-02, CEC-03, CEC-05, CEC-06, CEC-07, CEC-08, and CEC-10) and comes in third on the remaining two functions (CEC-04 and CEC-09). Furthermore, when analyzing the statistical standard deviation (SD) results, IGWO ties with CS as the most robust algorithms. Overall, IGWO demonstrates strong performance across the CEC 2019 function set.

## Judgment criteria

To evaluate the predictive capabilities of each model, we employed four metrics: mean square error (RMSE), mean absolute error (MAE), and coefficient of determination ($R^2$). These metrics are computed in Eqs. (39) to (41):

$$e_{RMSE} = \sqrt{\frac{1}{n} \sum_{1}^{n} (\hat{y}_i - y_i)^2} \tag{39}$$

$$e_{MAE} = \frac{1}{n} \sum_{1}^{n} |\hat{y}_i - y_i| \tag{40}$$

**Table 3 CEC2019 test suit.**

| Func | Descriptions | Dim | Range | $f_{min}$ |
|---|---|---|---|---|
| CEC-01 | Storn's Chebyshev polynomial fitting problem | 9 | [−8,192, 8,192] | 1 |
| CEC-02 | Inverse Hilbert matrix problem | 16 | [−16,384, 16,384] | 1 |
| CEC-03 | Lennard-Jones minimum energy cluster | 18 | [−4, 4] | 1 |
| CEC-04 | Rastrigin's function | 10 | [−100, 100] | 1 |
| CEC-05 | Griewangk's function | 10 | [−100, 100] | 1 |
| CEC-06 | Weierstrass function | 10 | [−100, 100] | 1 |
| CEC-07 | Modified Schwefel's function | 10 | [−100, 100] | 1 |
| CEC-08 | Expanded Schaffer's F6 function | 10 | [−100, 100] | 1 |
| CEC-09 | Happy Cat function | 10 | [−100, 100] | 1 |
| CEC-10 | Ackley function | 10 | [−100, 100] | 1 |

**Table 4 Statistical results of EWOA and other comparative algorithms on classical benchmark functions.** The models with bold formatting represent the best-performing ones.

| Functions | Indicator | Algorithms | | | | |
|---|---|---|---|---|---|---|
| | | IGWO | GWO | CS | ZOA | WOA |
| CEC-01 | Mean | **1.00E+00** | 7.51E+03 | 1.39E+04 | **1.00E+00** | 3.21E+06 |
| | Std Dev | **0.00E+00** | 1.92E+04 | 7.69E+03 | **0.00E+00** | 4.11E+06 |
| CEC-02 | Mean | **4.99E+00** | 1.06E+02 | 2.17E+02 | 5.00E+00 | 7.42E+03 |
| | Std Dev | 9.15E−06 | 5.26E+01 | 5.23E+01 | **0.00E+00** | 2.16E+03 |
| CEC-03 | Mean | **1.99E+00** | 2.61E+00 | 2.25E+02 | 7.27E+00 | 3.98E+00 |
| | Std Dev | **7.21E−01** | 1.81E+00 | 5.99E+01 | 7.30E−01 | 1.95E+00 |
| CEC-04 | Mean | 5.61E+01 | 7.37E+01 | **2.28E+01** | 1.40E+04 | 4.28E+01 |
| | Std Dev | 2.09E+01 | 1.72E+02 | **3.59E+00** | 5.70E+03 | 1.61E+01 |
| CEC-05 | Mean | **1.15E+00** | 1.32E+00 | 1.21E+00 | 5.58E+00 | 1.93E+00 |
| | Std Dev | **9.00E−02** | 2.10E−01 | 3.01E−01 | 9.02E−01 | 4.99E−01 |
| CEC-06 | Mean | **7.05E+00** | 1.01E+01 | 9.27E+00 | 1.07E+01 | 7.89E+00 |
| | Std Dev | **6.40E−01** | 8.10E−01 | 7.50E−01 | 8.84E−01 | 1.97E+00 |
| CEC-07 | Mean | **1.80E+01** | −2.74E+01 | −1.54E+02 | 9.32E+02 | 1.18E+03 |
| | Std Dev | 1.25E+02 | 1.27E+02 | **1.63E−02** | 3.15E+02 | 3.29E+02 |
| CEC-08 | Mean | **1.00E+00** | 1.00E+00 | **1.00E+00** | 1.55E+00 | 4.45E+00 |
| | Std Dev | 5.43E−08 | 4.00E−03 | **8.64E−15** | 1.85E−01 | 3.56E−01 |
| CEC-09 | Mean | 1.47E+00 | 4.97E+00 | 1.28E+00 | 6.49E+02 | **1.11E+00** |
| | Std Dev | 1.80E−01 | 1.67E+00 | **6.00E−02** | 2.18E+02 | 1.92E−01 |
| CEC-10 | Mean | **1.80E+01** | 2.09E+01 | 2.11E+02 | 2.14E+01 | 2.11E+01 |
| | Std Dev | 2.68E−01 | 2.47E+00 | 5.10E−01 | **1.06E−01** | 1.21E+00 |

Note:
IGWO, Improved Grey Wolf Optimizer; GWO: Grey Wolf Optimizer; CS: Cuckoo Search; ZOA: Zebra Optimization Algorithm; WOA: Whale Optimization Algorithm.

$$R^2 = \frac{\sum_1^n \left(\hat{y}_i - \bar{y}\right)^2}{\sum_1^n \left(y_i - \bar{y}\right)^2} \tag{41}$$

In financial time series forecasting, RMSE, MAE, and $R^2$ are employed as evaluation metrics because they collectively provide a comprehensive reflection of the model's predictive performance. Specifically, RMSE calculates the square root of the average of the squared prediction errors, making it more sensitive to larger errors and thus suitable for assessing the model's ability to predict extreme values. MAE, on the other hand, directly computes the average of the absolute differences between predicted and actual values, offering an intuitive measure of the average deviation of the predictions. Lastly, $R^2$ measures the proportion of the variance in the dependent variable that can be explained by the independent variables, indicating the model's fit to the data. The combined use of these three indicators allows for a thorough evaluation of the model's prediction accuracy and reliability.

## Model parameter selection

To enhance computational performance and accommodate large-scale data processing, we selected the Adam optimizer and set the number of wolves in the pack to 6. We utilized the Improved Grey Wolf Optimizer to optimize five hyperparameters: the time step, dropout rate, and the number of neurons in each of the three hidden layers. The search range varies based on the upper and lower limits of the parameters, and the dimensionality of the search space changes with the number of hyperparameters. We use the Mean Squared Error (MSE) of the prediction results as the fitness value. By navigating through the search space, the IGWO can identify the combination of hyperparameters that yields the lowest loss, which represents the optimal solution to the problem.

To avoid potential issues during the search process, such as excessive resource consumption due to an overly broad search range, we analyzed relevant research articles (*Hochreiter & Schmidhuber, 1997*; *Greff et al., 2016*; *Zhou et al., 2025*) that utilized LSTM models for predicting the Chinese financial derivatives market, as well as other related studies. The parameters employed in these articles have been tested through practice. Therefore, the parameter settings in these studies provide valuable guidance for us in setting our search range.

We observed that most researchers set the learning rate within the range of 0.001 to 0.01, with only a few opting for values outside this range. Additionally, many studies kept the number of LSTM layer neurons below 100 and the time step under 20. Furthermore, the number of epochs was typically set within 100 or around 200, with fewer studies choosing a higher number. The learning rate is set to 0.01, with the number of neurons in the LSTM layer ranging from 1 to 200. The DENSE layer contains 100 neurons, and the time steps range from 5 to 20. After testing, we observed negligible improvements in experiments with epochs exceeding 100. Considering environmental constraints, we limited the search range for epochs to 100. Simultaneously, boundary constraints have been incorporated. When a boundary is exceeded, a new position within the search space is

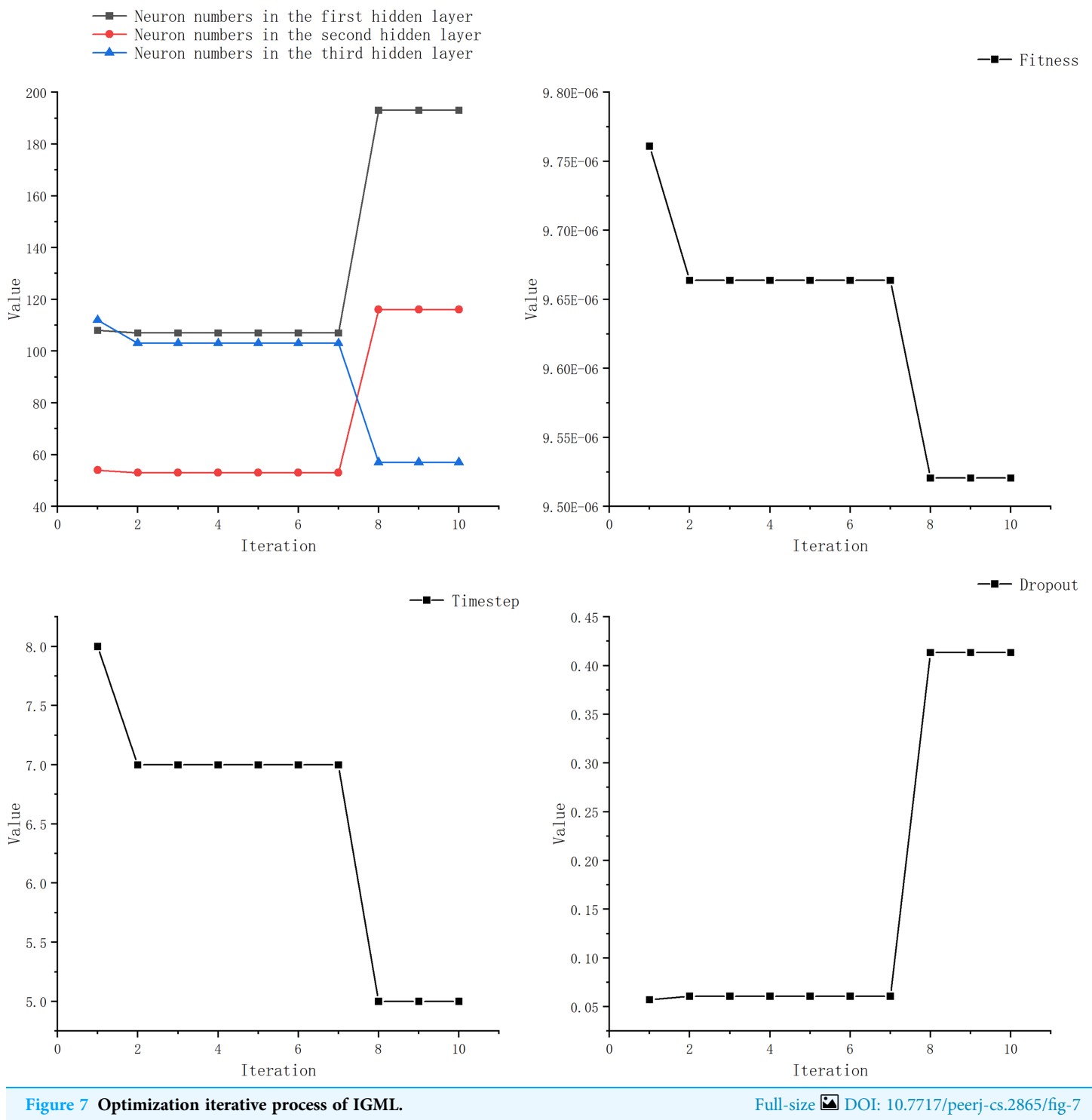

**Figure 7 Optimization iterative process of IGML.**

randomly selected. Once the search range for optimizing the target parameters has been defined, we can employ the GWO to determine the optimal parameters. Figure 7 and Table 5 illustrate the optimization process for each parameter.

**Table 5 Optimization iterative process of IGML.**

| Algorithm iterations | Neuron numbers in the first LSTM layer | Neuron numbers in the second LSTM layer | Neuron numbers in the LSTM hidden layer | Time step | Dropout | Fitness value |
|---|---|---|---|---|---|---|
| 1 | 108 | 54 | 112 | 8 | 0.05 | 9.76E−6 |
| 2 | 107 | 53 | 113 | 7 | 0.06 | 9.66E−6 |
| 3 | 107 | 53 | 113 | 7 | 0.06 | 9.66E−6 |
| 4 | 107 | 53 | 113 | 7 | 0.06 | 9.66E−6 |
| 5 | 107 | 53 | 113 | 7 | 0.06 | 9.66E−6 |
| 6 | 107 | 53 | 113 | 7 | 0.06 | 9.66E−6 |
| 7 | 107 | 53 | 113 | 7 | 0.06 | 9.66E−6 |
| 8 | 193 | 116 | 57 | 5 | 0.41 | 9.52E−6 |
| 9 | 193 | 116 | 57 | 5 | 0.41 | 9.52E−6 |
| 10 | 193 | 116 | 57 | 5 | 0.41 | 9.52E−6 |

**Table 6 Results of hyperparametric optimization.**

| Parameter | Search range | Optimal value |
|---|---|---|
| Neuron numbers in the first hidden layer | [1, 200] | 193 |
| Neuron numbers in the second hidden layer | [1, 200] | 53 |
| Neuron numbers in the third hidden layer | [1, 200] | 113 |
| Time step | [5, 20] | 5 |
| Dropout | [0.02, 0.5] | 0.41 |

From Table 6, it can be seen that the optimal parameters obtained by IGWO for LSTM are a dropout rate of 0.41, LSTM layer neuron numbers of 193, 53, and 113, and a time step of 5. Based on these optimized parameters obtained through IGWO, we constructed a prediction model for price forecasting. The neuron counts for the traditional models used for comparison were all set to 100, and the learning rate was set to 0.01.

## Experimental results and analysis

This section analyzes the predictive ability of IGML for price trends. Classic models such as BP, multi-layer perceptron (MLP), RNN, gated recurrent unit (GRU), LSTM, and bidirectional long short-term memory (BiLSTM) are widely used in futures price-spread prediction. We used these price-spread prediction models as comparison models for IGML in our experiments to directly test the effectiveness of the proposed model. Additionally, we reproduced the GWO algorithm to optimize the LSTM model and used GWO-LSTM as a comparative model in this study. By testing IGML's predictive ability, we can further compare the effectiveness of the IGWO and GWO population-based intelligent algorithms in optimizing the LSTM model.

Figure 8 illustrates that the classic MLP network can roughly predict price-spread trends. However, we find that when the futures price-spread fluctuates significantly, the predictive performance of MLP is poor, with a significant error between the predicted and

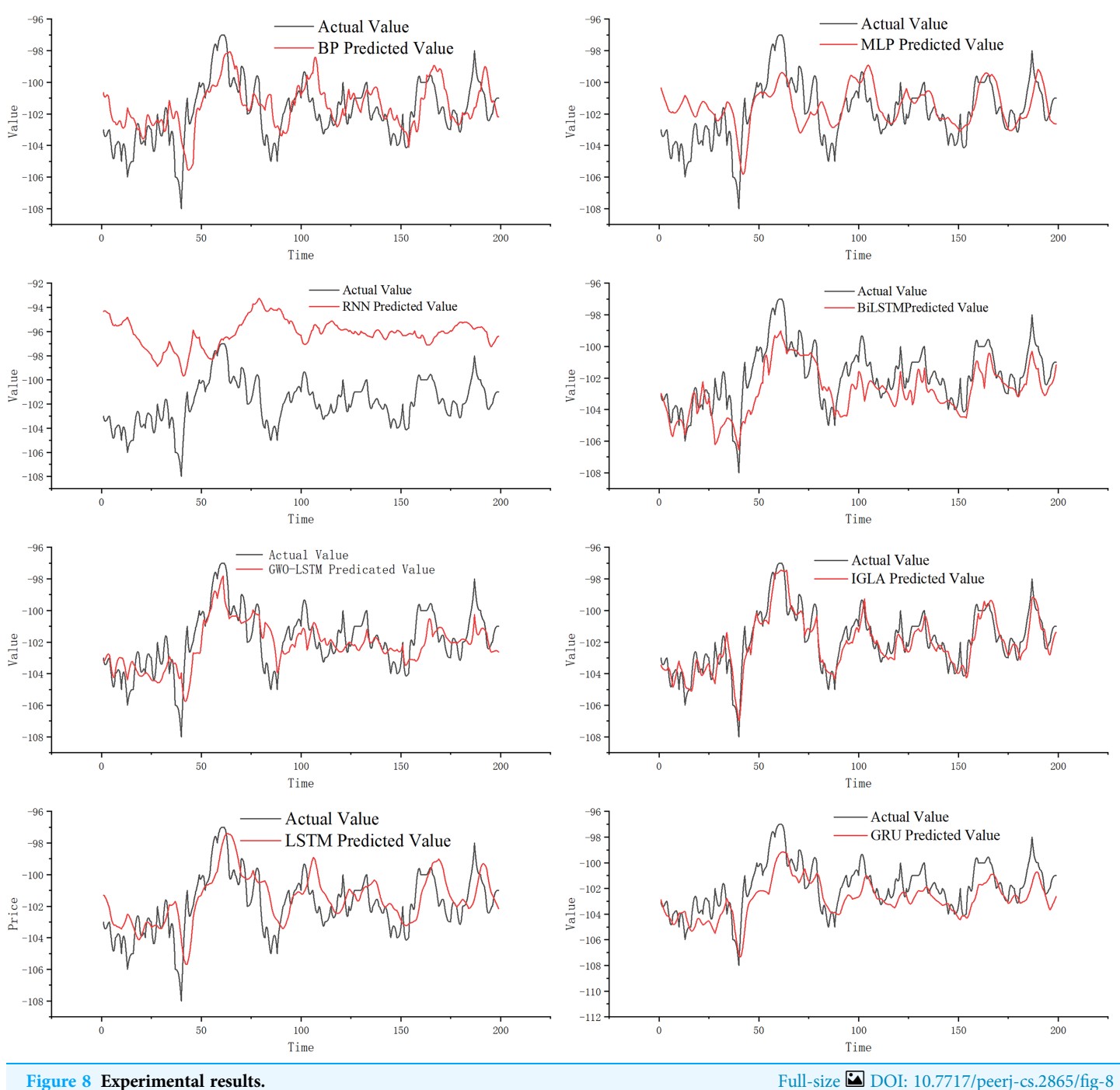

**Figure 8 Experimental results.**

actual curves. BP performs better than MLP, but the error is still considerable. The RNN model, known for handling time series problems, performs the worst on this dataset.

The LSTM network, a classic model for addressing time series problems, mitigates the issue of vanishing gradients. Observing the prediction curve, it is apparent that LSTM has enhanced its predictive performance, and the prediction errors have been reduced. An

**Table 7 Evaluation results of models.** The models with best-performance are indicated in bold.

| Model | RMSE | MAE | $R^2$ |
|---|---|---|---|
| BP | 2.7902 | 2.0558 | 0.9876 |
| MLP | 2.8319 | 2.1207 | 0.9794 |
| RNN | 11.6037 | 10.6687 | 0.8933 |
| GRU | 2.3610 | 1.7004 | 0.9604 |
| LSTM | 2.4001 | 1.8210 | 0.9825 |
| BILSTM | 2.2943 | 1.6490 | 0.9863 |
| GWO-LSTM | 2.0504 | 1.5069 | 0.9912 |
| IGML | **1.7313** | **1.2472** | **0.9955** |

improved version of LSTM, known as GRU, demonstrates even better prediction performance. Additionally, BiLSTM, another refined model based on LSTM, further diminishes prediction lag while maintaining high prediction accuracy. Although BiLSTM's fitting ability has significantly improved compared to LSTM, the prediction at turning points is still not ideal.

GWO-LSTM utilizes swarm intelligence optimization algorithms to fine-tune its model parameters, and we have found that the combined model exhibits improved prediction performance. And on this basis, IGWO further enhances the model performance significantly, resulting in IGML's superior prediction effect compared to GWO-LSTM. Compared to other prediction models, the IGML model demonstrates the smallest gap between its predicted values and the actual values, and it is the least affected by data fluctuations. This indicates that the IGML model possesses better fitting ability and risk resistance compared to other prediction methods. By analyzing the prediction curves, the conclusion can be more intuitively verified.

For a more intuitive and accurate comparison of the predictive performance among models, to highlight the superiority of the IGML model, the evaluation metrics for each model are shown in Table 7.

As shown in Table 7, MLP is an early model used in price-spread prediction, with values of 2.8319, 2.1207, and 0.9794 for its three evaluation metrics. Higher RMSE and MAE values indicate more significant prediction errors for MLP, while a smaller R-squared ($R^2$) value suggests weaker fitting capability. BP shows a significant improvement in prediction compared to MLP, with RMSE, MAE, and $R^2$ values of 2.7902, 2.0558, and 0.9876, respectively. However, BP's predictions are still not accurate enough based on the prediction curve and evaluation metrics. RNN exhibits a sharp decline in performance with RMSE, MAE, and $R^2$ values of 11.6037, 10.6687, and 0.8933, respectively, indicating its unsuitability for this dataset compared to other models.

LSTM, as a classic method in price-spread prediction, shows a significant enhancement in prediction capability after addressing the gradient vanishing problem of RNN. Its metrics are 2.4001, 1.8210, and 0.9825, indicating some errors compared to the proposed model. GRU, an optimized model based on LSTM, outperforms LSTM in error metrics (RMSE and MAE) with values of 2.3610 and 1.7004, respectively, but has a lower $R^2$ value

than LSTM. BiLSTM, another optimized model based on LSTM, shows lower error metrics (RMSE and MAE) than LSTM and a higher $R^2$ value.

By incorporating swarm intelligence algorithms with LSTM, the GWO-LSTM model demonstrates enhanced prediction capabilities, with metric values of 2.0504, 1.5069, and 0.9912 respectively. These metric values significantly outperform those of traditional prediction methods, validating the effectiveness of using swarm intelligence algorithms to optimize LSTM model parameters.

The proposed IGML model, using IGWO to optimize LSTM, further enhances prediction results compared to GWO-LSTM as expected. The metric values for IGML are 1.7313, 1.2472, and 0.9955, showing a 37.9%, 38.8%, 85.1%, 26.6%, 27.8%, 24.5%, and 15.5% lower RMSE compared to BP, MLP, RNN, GRU, LSTM, BiLSTM, and GWO-LSTM models, respectively. The MAE metric also decreases by 39.3%, 41.1%, 88.3%, 26.6%, 31.5%, 24.3%, and 17.2% compared to these models. These metrics indicate that IGML has smaller errors between predicted and actual values, thus higher prediction accuracy than other models.

In Table 7, IGML's $R^2$ value is closest to 1 among all models, indicating strong fitting capability. Overall, these experimental results demonstrate that IGML has the best predictive capability and effectiveness compared to other methods.

Based on the above analysis, the IGML model is more effective and has higher prediction accuracy in handling price-spread time series data compared to traditional neural network models. Specifically, when dealing with complex time series data, the IGML model is able to more accurately capture the dynamic changes and underlying patterns in the data, thereby generating more precise prediction results. These advantages make the IGML model highly valuable and competitive in financial applications such as futures spread prediction.

## Backtesting experiment

To validate the performance of IGML in actual trading, this article conducts a backtesting comparison using three models. We utilized correlated data from rebar and hot-rolled coil, with the rebar 5-min K-line data used for model training. The backtesting was then performed on the hot-rolled coil 5-min K-line data from 2019 to 2024, using three different models: Rbreaker, Rbreaker-LSTM-Attention (Rbreaker-LA), and Rbreaker-IGML. In the Rbreaker-LA strategy, we introduced the LSTM-Attention model to predict market trends in terms of rises and falls. This prediction served as Supplemental Information to assist the Rbreaker strategy in making more precise trading decisions. The Rbreaker-IGML strategy further optimized the Rbreaker-LA by incorporating the Improved Grey Wolf Optimizer (IGWO) to tune the hyperparameters within the strategy. Specifically, we adjusted key parameters in the formulas for breakthrough buy price, observation sell price, reversal sell price, reversal buy price, observation buy price, and breakthrough sell price, aiming to achieve better trading performance. The results of the backtesting experiments have been compiled in the following table for further analysis and discussion.

**Table 8 Comparison of backtesting results.** The best performance models are shown in bold.

| Indicator | Rbreaker | Rbreaker-LA | Rbreaker-IGML |
|---|---|---|---|
| Initial capital | 50,000 | 50,000 | 50,000 |
| Ending capital | 61,772.3 | 71,965.3 | **168,233.7** |
| Total profit/loss | 11,772.3 | 21,965.3 | **118,233.7** |
| Average profit/loss | 60.4 | **264.6** | 185.6 |
| Return rate | 23.3% | 43.9 | **212.3%** |
| Annualized return rate | 4.61% | 8.78% | **42.6%** |
| Total number of trades | 195 | 83 | 637 |
| Total number of profitable trades | 118 | 50 | 387 |
| Average profit | 214.4 | 839.5 | 1,118.8 |
| Maximum profit | 1,217.1 | 3,955.1 | 12,121.1 |
| Number of losing trades | 77 | 33 | 250 |
| Average loss | −175.6 | −606.4 | −1,259.0 |
| Maximum loss | −1,507.4 | −2,595.9 | −7,537.1 |
| Maximum drawdown ratio | **0.39%** | 1.38% | 3.66% |
| Maximum drawdown amount | **197.46** | 669.98 | 2,209.66 |
| Sharpe ratio | −1.9853 | −0.34 | **1.12** |

Based on the data presented in Table 8, we can clearly see that the Rbreaker algorithm exhibits an advantage in the metric of maximum drawdown, with the smallest value among all. This indicates that the Rbreaker algorithm performs exceptionally well in risk control, effectively reducing potential losses during the investment process. When we combine the LSTM-Attention mechanism with the Rbreaker algorithm, it can be observed that this integration improves the strategy's return rate and Sharpe ratio at the cost of increased drawdown. The increase in return rate signifies higher investment returns, while the improvement in Sharpe ratio indicates that the strategy achieves higher excess returns for the same level of risk. In this article, we propose the combination of the IGML model with the Rbreaker algorithm. After adjusting the parameters using IGWO, although the new strategy slightly underperformed the previous two in terms of maximum drawdown, it achieved significant improvements in both return rate and Sharpe ratio. The results indicate that through reasonable algorithm integration and parameter optimization, not only is good risk control capability maintained, but the profitability and risk-adjusted returns of the investment strategy are further enhanced. Furthermore, parameter optimization is not only applicable to optimizing model hyperparameters, but can also be used to optimize parameters in financial trading strategies, demonstrating excellent adaptability.

## CONCLUSION

This study introduces an innovative IGML model for futures spread prediction, which integrates LSTM networks with an IGWO. By optimizing the hyperparameters of LSTM through the IGWO algorithm, the model has made significant progress, including establishing an objective criterion for architecture selection, which reduces human bias

compared to manual tuning. Cross-validation on two highly correlated steel futures contracts from the Shanghai Futures Exchange demonstrates the model's excellent generalization ability. The experimental validation was conducted using 1-min high-frequency K-line spread data from the Shanghai Futures Exchange, along with a backtesting comparison experiment. The results show that the IGML model outperforms traditional machine learning methods in terms of directional accuracy and error reduction, demonstrating significant practical value. For instance, automated parameter optimization reduces model development time, enhanced prediction stability provides more reliable guarantees for generating trading signals, and the framework can be transplanted for spread trading across related commodity pairs. Furthermore, the IGML model can also be applied to other types of financial data, such as stocks or cryptocurrencies, by incorporating domain-specific characteristics and leveraging advanced machine learning techniques. By tailoring input features and model architecture according to the characteristics of different financial data, the IGML model can provide accurate and timely predictions for a wide range of financial assets.

This model holds practical significance for both traders and policymakers. For traders, it enables them to gain a more precise and nuanced understanding of market behavior, thereby aiding them in making more informed decisions about when to buy, sell, or hold assets. By incorporating complex factors such as economic indicators, sentiment analysis, and historical trends, the model can reveal patterns and insights that may be overlooked by traditional methods. For policymakers, this model serves as an invaluable tool for forecasting economic trends and assessing the impact of various policies. By simulating different scenarios, policymakers can gain deep insights into how changes in interest rates, fiscal policies, or regulations might affect markets and the broader economy, allowing for more effective and targeted interventions.

However, despite these achievements, there are still limitations that require further research. The dependency on high-frequency data may limit the model's applicability in illiquid markets, and the current implementation primarily focuses on pairwise spreads rather than complex multi-asset portfolios. In addition, computational efficiency needs to be optimized for real-time trading systems. To address these limitations and advance the field, future research directions should focus on expanding the framework to incorporate macroeconomic indicators to accommodate different market environments, implementing online learning protocols to adapt to dynamic market conditions, and conducting real-time trading experiments to quantify economic value creation.

In future research, the IGML model can be further improved in several aspects. Firstly, more advanced machine learning algorithms and deep learning architectures can be explored to enhance the model's predictive accuracy and generalization capability. This includes trying different feature selection methods, optimizing model parameters, and incorporating more contextual information to improve the model's understanding. Additionally, research can be conducted on how to integrate more types of financial data, such as news and social media sentiment, into the model to provide a more comprehensive market analysis.

In summary, this study has built a bridge between computational intelligence and financial engineering, offering contributions in both methodology and practice. It provides a scalable foundation for algorithmic trading. With further research and optimization, this model has the potential to aid in the analysis and trading of financial markets.

### Funding

This work was funded by Henan Provincial Key R&D Special Project "Key Technology and Industrialisation of Intelligent Fusion of Multi-Source Heterogeneous Sensors Based on New Generation Communication Technology" (No. 231111210500), Henan University Science and Technology Innovation Team Support Plan (No. 20IRTSTHN013) and Henan Province Key R&D and Promotion Special Project (No. 212102210166). The Henan Province Key R&D funded the APC for this article. The funders had no role in study design, data collection and analysis, decision to publish, or preparation of the manuscript.

### Grant Disclosures

The following grant information was disclosed by the authors:
Henan Provincial Key R&D Special Project: 231111210500.
Henan University Science and Technology Innovation Team Support Plan: 20IRTSTHN013.
Henan Province Key R&D and Promotion Special Project: 212102210166.

### Competing Interests

The authors declare that they have no competing interests.

### Author Contributions

- Yongli Tang performed the computation work, prepared figures and/or tables, authored or reviewed drafts of the article, and approved the final draft.
- Zhenlun Gao conceived and designed the experiments, performed the experiments, analyzed the data, prepared figures and/or tables, authored or reviewed drafts of the article, and approved the final draft.
- Zhongqi Cai performed the experiments, authored or reviewed drafts of the article, and approved the final draft.
- Jinxia Yu analyzed the data, prepared figures and/or tables, and approved the final draft.
- Panke Qin performed the computation work, prepared figures and/or tables, authored or reviewed drafts of the article, and approved the final draft.

### Data Availability

Code and raw data are available in the Supplemental Files.
The original data used in the analysis originated from the Shanghai Futures Exchange: https://www.shfe.com.cn/reports/tradedata/datadownload/.

## Supplemental Information

Supplemental information for this article can be found online at http://dx.doi.org/10.7717/peerj-cs.2865#supplemental-information.

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
