# Peer review of "Enhanced futures price-spread forecasting based on an attention-driven optimized LSTM network: integrating an improved grey wolf optimizer algorithm for enhanced accuracy"

_PeerJ Computer Science, doi:10.7717/peerj-cs.2865_

## Round 0.1 · original submission · Major Revisions

Dear authors,


Reviewers have now commented on your article. We do encourage you to address the concerns and criticisms of the reviewers with respect to reporting, experimental design, and validity of the findings and resubmit your article once you have updated it accordingly. Following should also be addressed:

1. A major criticism for the current version of the paper is the lack of justification for the use of the Grey Wolf Optimization algorithm as the optimization method for the model. Therefore, you need to add a standard evolutionary algorithm as an additional point of comparison for your ABC algorithm.
2. Configuration space of evolutionary algorithms should be detailed. It should be more specific and comprehensive. Representation scheme (encoding type) and fitness function with constraint functions should be clearly provided. How constraints (for example: for decision variables) are handled should also be provided.
3. Name of the optimization method is not consistent. “grey wolf” or “gray wolf”? In the title you have used “Gray Wolf”, however in the Abstract you have used “Grey Wolf”.
4. Keywords in the Abstract section should be correctly listed according to the alphabetical order.
5. Please pay special attention on the usage of abbreviations.
6. “this paper.” In line 141 should be corrected.
7. Many of the equations are part of the related sentences. Attention is needed for correct sentence formation.
8. Equations should be used with correct equation number. Please do not use “as follows”, “given as”, etc. Explanation of the equations should also be checked. All variables should be written in italic as in the equations. Their definitions and boundaries should be defined. Necessary references should be provided.
9. All of the values for the parameters of all algorithms should be given.
10. It is recommended that the paper's experimental results be discussed in greater depth, with additional recommendations and conclusions provided. The conclusion section is lacking in several respects. Firstly, it is essential to describe the academic implications, main findings, shortcomings and directions for future research. Secondly, the conclusion is currently confusing. It is necessary to clarify what will happen next and what we should expect from future papers. To address these issues, the conclusion should be rewritten, taking the following comments into consideration:
- Highlight your analysis and reflect only the important points for the whole paper.
- Mention the benefits
- Mention the implication in the last of this section.

Best wishes,

·

Basic reporting

What are the main challenges of financial market prediction?
How do time-series characteristics of futures price data complicate prediction tasks?
Why is it essential to mine and fuse multiple heterogeneous data in financial markets?
How does Long Short-Term Memory (LSTM) help in capturing long-term dependencies in data?
What are the limitations of traditional machine learning methods for financial market prediction?

Experimental design

How does the Multi-Head Self-Attention mechanism improve the focus on crucial features in LSTM?
Why is hyperparameter optimization critical for the performance of LSTM models?
What is the role of the Grey Wolf Optimization (GWO) algorithm in the IGML model?
What are the four improvement strategies proposed for the GWO algorithm?
How does the Improved Grey Wolf Optimizer (IGWO) outperform the standard GWO algorithm?
Why was the futures price-spread dataset chosen for this research?
What preprocessing steps were applied to the dataset before using the IGML model?
How was the real futures price-spread dataset structured for experimentation?
What benchmark problems were used to test the effectiveness of the IGWO algorithm?
How were the experiments designed to compare IGWO with conventional algorithms?

Validity of the findings

Why are RMSE and MAE suitable metrics for evaluating the IGML model's performance?
How does the IGML model achieve up to 88% improvement in RMSE?
What factors contribute to the 85% enhancement in MAE achieved by the IGML model?
How do the performance metrics of IGML compare to traditional financial prediction models?
What insights can be drawn from the RMSE and MAE improvements in futures spread forecasting?

Additional comments

How does IGML differ from traditional LSTM-based models?
What are the advantages of the IGML model over other hyperparameter optimization techniques?
How does the inclusion of the Multi-Head Self-Attention mechanism improve the prediction capabilities?
In what ways does IGWO outperform conventional optimization algorithms?
How do the improvements in GWO affect the overall performance of the IGML model?
What are the key features of the Multi-Head Self-Attention mechanism used in the model?
How do the proposed improvements to the GWO algorithm enhance its optimization capabilities?
What specific aspects of the IGWO make it more effective for hyperparameter tuning?
How does the IGML model balance computational efficiency with prediction accuracy?
What challenges were encountered in integrating IGWO with the LSTM model?
How can the IGML model be applied to other financial forecasting tasks?
What are the practical benefits of using IGML for futures price-spread prediction?
How might the IGML model influence decision-making for investors and policymakers?
What potential does the IGML model hold for real-time financial forecasting?
How does the IGML model contribute to the broader field of AI in finance?
What potential improvements can be made to the IGML model in future research?
How might the IGML model be adapted to other types of financial data, such as stocks or cryptocurrencies?
Could the IGML framework be integrated with other deep learning architectures, such as Transformers?
What additional datasets could be used to further validate the effectiveness of the IGML model?
How might the IGML model evolve to handle even more complex financial prediction tasks?

·

Basic reporting

1. Simplify the abstract to focus on the problem, proposed solution, and key results. Remove less critical details for a stronger impact.
2. The introduction should elaborate more on the limitations of existing models in futures price-spread forecasting and how they fail to address specific challenges.
3. Expand the related works to include recent advancements in metaheuristic optimization techniques and their applications in financial forecasting.

Experimental design

1. Contributions are clear but could benefit from elaboration on how IGML distinctly outperforms similar models.
2. Include a discussion on why the selected metrics (RMSE, MAE, R²) are particularly relevant for this study.
3. Elaborate on the rationale behind the chosen search ranges for hyperparameters.
4. Provide more details about the preprocessing steps for the Shanghai Futures Exchange data and justify the choice of features.

Validity of the findings

1. Address how the model performs with larger datasets or in different financial domains beyond futures spreads.
2. Discuss the potential practical implications of this model for traders or policymakers.
3. Comparison with additional state-of-the-art techniques could strengthen the results.
4. While improvements in prediction accuracy are noted, the computational overhead of the IGWO is not adequately addressed.

Additional comments

1. The use of trigonometric functions for convergence factors is innovative but needs more theoretical justification or citations.
2. Citations are relevant but could include more recent papers, particularly on self-attention mechanisms and hybrid optimization techniques.
3. Include the limitations of the proposed model.
4. Some minor grammatical issues and inconsistencies in figures/tables formatting should be addressed.

Reviewer 3 ·

Basic reporting

overall the paper is okay.

Experimental design

Experimental design has flaws as explained in 4.

Validity of the findings

Not valid due to design flaws as explained in 4.

Additional comments

The paper aims to predict futures price-spread. The paper's main claim is that their Improved Grey Wolf Optimizer results in lower RMSE/MAE.
1. I claim the overall purpose of predicting spread (or prices) to make profit from financial markets, however, authors have not shown whether their improvements result in making profits from the markets. I argue MAE and RMSE are wrong metrics to be used in this domain. MAE and RMSE are measures of error, not profit. Even if a model has a low RMSE, this does not necessarily mean that it will generate profits in the market. For example, a model might accurately predict the spreads (or direction of stock prices) but fail to generate profits because it does not take into account transaction costs, liquidity constraints, slippage or other factors that can affect real-world trading. Moreover, RMSE assumes that the distribution of errors is normal, which is not the case in financial markets. In real-world trading, spreads can be highly volatile and subject to sudden, unexpected movements due to news that can cause significant losses even if the model's predictions are accurate on average.
2. The correlation coefficient presented which again has no relation with the profitability of the algorithm unless the authors prove otherwise. R2 improvements is very small, moreover.
3. No backtesting was performed, as an example in this paper https://doi.org/10.3390/asi4010017 authors have provided backtesting analysis and the potential profitability of their algorithm. Generally, it is argued that spreading your investment across various instruments in a financial market will double your investment every 10 years. Therefore, algorithms must outperform this metric in order for them to be considered useful algorithms.
4. So many other papers have been published claiming their algorithm is better in this domian, therefore, authors should compare their results against previously published studies and establish that their methods make more money.

---

## Round 0.2 · Major Revisions

Dear Authors,

It is evident that there have been improvements to the quality of the paper; however, a further review is required in order to ensure that all issues have been addressed. We encourage you to address the concerns and criticisms that are listed below.

Best wishes,

·

Basic reporting

What are the primary challenges faced in financial market prediction, particularly in futures price-spread data?
Why do traditional machine learning methods struggle with mining patterns in financial market data?
What limitations do conventional LSTM models have in financial market prediction?
How does the IGML model address the shortcomings of traditional LSTM models?
What role does the multi-head self-attention mechanism play in the IGML model?
What are the four strategic enhancements introduced in the Improved Grey Wolf Optimizer (IGWO)?
How is IGWO validated in terms of optimization performance?
What performance improvements does the IGML model achieve compared to baseline models?
What evaluation metrics are used to assess the IGML model’s performance?
What are the key contributions of this paper to the field of financial market prediction?

Experimental design

What are the key characteristics of futures price-spread data that make prediction challenging?
How does heterogeneous data affect financial market predictions?
What specific hyperparameters in LSTM models require optimization for better performance?
Why is feature prioritization important in financial market forecasting?
How does the multi-head self-attention mechanism improve feature interaction?
What distinguishes IGWO from the standard Grey Wolf Optimizer (GWO)?
What types of optimization problems were used to validate IGWO’s convergence efficiency?
How does IGWO enhance the process of automated hyperparameter tuning?
What datasets were used for evaluating the IGML model’s performance?
What baseline models were used for comparison in the study?
How does the IGML model handle the complex temporal dependencies in financial data?
What are the key advantages of self-attention mechanisms over traditional recurrent layers?
What are the four strategic enhancements introduced in IGWO, and how do they improve optimization?
What machine learning or deep learning techniques were considered but not used in this study?
How does RMSE (Root Mean Square Error) reflect the performance of financial forecasting models?
Why is MAE (Mean Absolute Error) used as a performance metric in this research?
How does the IGML model compare to standard LSTM and Transformer-based models?
What are some potential limitations of the IGML model?
Could the IGML model be applied to other financial forecasting tasks beyond futures price-spread data?

Validity of the findings

What impact does hyperparameter tuning have on model performance in time series forecasting?
How does IGWO compare to other hyperparameter optimization techniques such as Bayesian Optimization or Genetic Algorithms?
What are some real-world applications where the IGML model could be beneficial?

Additional comments

What computational resources are required to train and optimize the IGML model?
How does the IGML model balance computational efficiency with predictive accuracy?
What challenges might arise when implementing IGML in live financial trading environments?
How could the IGML model be further improved in future research?
What are the implications of this research for financial analysts and investors?
Does the paper discuss any risks associated with using AI-based financial prediction models?
How does the research contribute to the broader field of financial market analysis?
What future directions does the paper suggest for improving financial forecasting models?

·

Basic reporting

The quality of this research has significantly improved after the revisions.

Experimental design

The experimental design of this study has been significantly improved after the revisions, ensuring a more rigorous and reliable evaluation of the proposed methodology.

Validity of the findings

The validity of the findings in this study has been significantly strengthened after the revisions.

Reviewer 3 ·

Basic reporting

no comment

Experimental design

no comment

Validity of the findings

1) Authors have performed back testing (rebuttal letter) to show the profitability of their algorithm however they did not include these results in the manuscript.

2) The authors should add a new section in the Results section and show profitability on unseen data that is not used for the training. For example, they should use out-of-sample earlier data for training like 2019 to 2021 and then then use 2022 to 2024 for backtesting. Their training and backtesting data overlapped which is wrong. Backtesting must be performed on the recent data without any overlap with training or test data used in machine/deep learning, since the algorithm is optimized on training and test data.

3) Line 336 Authors state 'The dataset is divided into a training set (80%), a test set (20%), and a validation set (10% of the training set). There is no mention of how the split of the data was done? Was it random or time-based or what?

---

## Round 0.3 · Minor Revisions

Dear Authors,

Please clearly address the concerns and criticisms of Reviewer 1.

Best wishes,

·

Basic reporting

Theoretical explanation given from my suggestions but not incorporated into the manuscript.


My suggestion should be to update in the manuscript, then resend review again

Experimental design

No explanation to all results images and tables.

Equations mentioned but not clear explanation

Validity of the findings

Theoretical explanation given from my suggestions but not incorporated into the manuscript.


My suggestion should be to update in the manuscript, then resend review again

Additional comments

Need to improve Manuscript

---

## Round 0.4 · accepted · Accept

Dear Authors,

Thank you for clearly addressing the reviewers' comments. Your paper seems sufficiently improved and ready for publication.

Best wishes,